# Depth Dose Enhancement in Orthovoltage Nanoparticle-Enhanced Radiotherapy: A Monte Carlo Phantom Study

**DOI:** 10.3390/mi14061230

**Published:** 2023-06-10

**Authors:** James C. L. Chow, Sama Jubran

**Affiliations:** 1Radiation Medicine Program, Princess Margaret Cancer Centre, University Health Network, Toronto, ON M5G 1X6, Canada; 2Department of Radiation Oncology, University of Toronto, Toronto, ON M5T 1P5, Canada; 3Department of Physics, Toronto Metropolitan University, Toronto, ON M5B 2K3, Canada; sama.jubran@torontomu.ca

**Keywords:** nanoparticles, radiosensitizer, radiation therapy, Monte Carlo simulation, cancer therapy, orthovoltage photon beam, skin therapy

## Abstract

Background: This study was to examine the depth dose enhancement in orthovoltage nanoparticle-enhanced radiotherapy for skin treatment by investigating the impact of various photon beam energies, nanoparticle materials, and nanoparticle concentrations. Methods: A water phantom was utilized, and different nanoparticle materials (gold, platinum, iodine, silver, iron oxide) were added to determine the depth doses through Monte Carlo simulation. The clinical 105 kVp and 220 kVp photon beams were used to compute the depth doses of the phantom at different nanoparticle concentrations (ranging from 3 mg/mL to 40 mg/mL). The dose enhancement ratio (DER), which represents the ratio of the dose with nanoparticles to the dose without nanoparticles at the same depth in the phantom, was calculated to determine the dose enhancement. Results: The study found that gold nanoparticles outperformed the other nanoparticle materials, with a maximum DER value of 3.77 at a concentration of 40 mg/mL. Iron oxide nanoparticles exhibited the lowest DER value, equal to 1, when compared to other nanoparticles. Additionally, the DER value increased with higher nanoparticle concentrations and lower photon beam energy. Conclusions: It is concluded in this study that gold nanoparticles are the most effective in enhancing the depth dose in orthovoltage nanoparticle-enhanced skin therapy. Furthermore, the results suggest that increasing nanoparticle concentration and decreasing photon beam energy lead to increased dose enhancement.

## 1. Introduction

Orthovoltage skin therapy is a type of radiotherapy that utilizes low-energy X-rays in the kilovoltage (kV) range to treat skin cancer and other dermatological conditions [1,2]. This technique is specifically designed to target the cancerous cells while minimizing the impact on healthy skin tissue. Orthovoltage skin therapy is a non-invasive procedure that delivers a high dose of radiation to the surface of the skin, making it an effective treatment option for superficial lesions. The therapy can be administered using a range of photon beam energies, which allows for personalized treatment based on the type and severity of the cancer. Overall, orthovoltage skin therapy has proven to be a safe and effective treatment for a variety of skin conditions [3,4].

Heavy metal nanoparticles, such as gold, platinum, and silver, are being investigated for their potential to enhance the effectiveness of orthovoltage radiotherapy [5,6,7]. When heavy metal nanoparticles are introduced into the tumor site, they can increase the absorption of radiation and amplify the dose delivered to the tumor cells. The reason for the high dose enhancement of heavy metal nanoparticles in kV photon beams is due to the relationship between the photoelectric effect cross section and the atomic number of the nanoparticle [8]. The cross section is directly proportional to the atomic number and inversely proportional to the photon beam energy. As heavy metal nanoparticles have a higher atomic number than other nanoparticles, they are more effective at absorbing radiation and generating secondary electrons when irradiated by kilovoltage photon beams. This leads to a higher dose of radiation delivered to the tumor cells and greater therapeutic efficacy compared to nanoparticles with lower atomic numbers and exposure to megavoltage photon beams [9]. Additionally, the unique physiochemical properties of heavy metal nanoparticles, including their high atomic numbers and strong absorption of X-rays, make them particularly suitable for use in orthovoltage radiotherapy [10]. Research in this field is ongoing, but early results have been promising and suggest that heavy metal nanoparticle-enhanced orthovoltage radiotherapy has the potential to be a safe and effective treatment option for cancer patients [11,12,13].

Monte Carlo simulation is a powerful tool that has been widely used in nanoparticle-enhanced radiotherapy research. This technique uses a mathematical algorithm to simulate the transport of radiation particles in a given medium and is particularly useful in predicting the dose distribution in complex biological systems [14]. In nanoparticle-enhanced radiotherapy, Monte Carlo simulation can be used to predict the dose enhancement ratio, which is the ratio of the dose delivered to a tissue with nanoparticles to the dose delivered without nanoparticles [15]. This information can help optimize the nanoparticle concentration and photon beam energy used in radiotherapy to achieve the greatest therapeutic benefit while minimizing damage to healthy tissues [16]. While it is true that the relationship between depth dose and variations in nanoparticle variables can be estimated to some extent using basic coefficients such as atomic number, attenuation coefficient, and particle interaction cross-section, the utilization of Monte Carlo simulation allows researchers to gain a deeper understanding of the underlying physical mechanisms driving nanoparticle-enhanced radiotherapy. This advanced method enables the design of more effective treatment strategies by providing valuable insights into the complex interactions involved. Moreover, while Monte Carlo simulation can be computationally intensive, recent advancements in computer hardware and software have made it more accessible and practical for use in clinical research [17,18].

Previous studies investigating dose enhancements in nanoparticle-enhanced radiotherapy have focused on depth dose enhancements using megavoltage (MV) photon beams, which are typically used to treat deep-seated tumors [19,20,21,22]. These studies have examined the impact of nanoparticle materials, concentration, and beam energy on depth dose enhancements for MV flattening-filter-free and flattening-free photon beams in various clinical scenarios. However, there is a lack of research on depth dose enhancements in orthovoltage nanoparticle-enhanced skin therapy, which uses kV photon beams. The goal of this study is to investigate the relationship between depth dose enhancement and treatment variables, such as nanoparticle materials, concentration, and photon beam energy, in nanoparticle-enhanced radiotherapy using kV photon beams. Monte Carlo simulation based on the macroscopic approach was employed to determine the depth dose from a water phantom [15].

## 2. Materials and Methods

### 2.1. Monte Carlo Simulation

The Electron Gamma Shower (EGSnrc) code, developed by the National Research Council of Canada, was employed in this study [23]. The X-ray energy spectra was improved by integrating the electron impact ionization model, while the directional bremsstrahlung splitting enhanced the energy transition efficiency from electron current to photons.

In this study, the BEAMnrc code [24] was utilized to model the 105 kVp and 220 kVp photon beams produced by a Gulmay D3225 X-ray machine. The maximum power of the kVp beam treatment machine was 3 kW with tube currents of 20 mA. To generate phase-space files, particles crossing a scoring plane at the bottom of the circular cone (5 cm diameter) were tracked for a source-to-surface distance (SSD) of 20 cm. The treatment head data of geometries and materials of different components were provided by the manufacturer. The beam qualities of the 105 kVp and 220 kVp beams were defined by specific filters to remove the low-energy photons. In the case of the 105 kVp beam, a 2.4 mm aluminum filter was utilized, whereas for the 220 kVp beam, a combination of a 1 mm aluminum filter and a 1.2 mm copper filter was used in the simulation. These filters were chosen to tailor the energy spectra and optimize the characteristics of the respective beams for the purposes of our study. The Monte Carlo simulations were verified by comparing the percentage depth doses in uniform water, calculated and measured by a parallel-plate ionization chamber (PS-033, Capintec, Ramsey, NJ, USA), and the results showed good agreement [11,25]. Rayleigh scattering was included in the simulation, and the energy cut-offs for the electron and photon transport were set to 521 keV and 1 keV, respectively [11,25].

The PEGS4 code based on EGSnrc was utilized to create material data sets for different nanoparticle concentrations [23]. Cross-section data for particle interaction with water were generated for concentrations of gold, platinum, iodine, silver, and iron oxide nanoparticles (3, 7, 18, 30, and 40 mg/mL). The nanoparticle concentration was selected in this study based on preclinical experiments [16]. It is important to acknowledge that while Zheng et al. [11] conducted a study using similar nanoparticle concentration and photon beam energy, their focus was on comparing the dose enhancement at a tumor site using kilovoltage photon and megavoltage electron beams in skin therapy. They explored the effects of different skin tumor thicknesses through simulations involving various energies of photon and electron beams [11]. In contrast, the emphasis of our study lies in examining the variations of depth dose associated with nanoparticle variables specifically in the context of kilovoltage photon beams utilized in skin therapy. In Figure 1 of the Monte Carlo simulation, a water phantom with dimensions of 10 × 10 × 12 cm^3^ was irradiated by 105 kVp and 220 kVp photon beams. The field size was 5 cm in diameter, and the SSD was 20 cm. The chosen simulation geometry was specifically designed to focus on the depth dose variations along the central beam axis while mitigating the potential impacts of other parameters such as phantom heterogeneity, beam quality, and angle. The phantom was composed of pure water or a mixture of water and nanoparticles at concentrations ranging from 3 to 40 mg/mL. The simulation utilized gold, platinum, iodine, silver, and iron oxide nanoparticle materials added to the water phantom. By varying the photon beam energy, nanoparticle material, and concentration, Monte Carlo simulation using the EGSnrc-based DOSXYZ code determined depth doses up to 10 cm along the central beam axis of the phantom. The macroscopic approach was employed in the simulation. By employing a substantial number of histories, specifically 200 million for each simulation [26,27], we were able to attain a statistical uncertainty of ±1% for the depth dose in our simulations. This number of histories proved effective in minimizing statistical variations and ensuring a higher level of accuracy in the obtained depth dose results.

### 2.2. Calculation of Dose Enhancement Ratio

To quantify the increase in depth dose enhancement resulting from the addition of nanoparticles, a dose enhancement ratio (DER) can be calculated [28]. This involves determining the radiation dose at a certain depth with nanoparticles added and comparing it to the dose at the same depth without nanoparticles (i.e., using water alone). The DER is expressed as:(1)$$DER=\frac{Depth\;dose\;at\;a\;point\;with\;addition\;of\;nanoparticles}{Depth\;dose\;at\;the\;same\;point\;without\;addition\;of\;nanoparticles}$$

Equation (1) shows that the DER is the ratio of the dose after nanoparticles are added to the dose before nanoparticles are added in the water phantom. As nanoparticles can increase the dose, it is expected that the DER value will increase with their addition. Therefore, DER values are expected to be greater than one. This study aimed to investigate the effect of photon beam energy, nanoparticle material, and concentration on the variation of DER values.

## 3. Results

Figure 2 displays the variation of DER values for 105 kVp photon beams against the nanoparticle concentration, plotted against the phantom depth. Gold, platinum, iodine, silver, and iron oxide nanoparticle materials were used in Figure 2a–e, respectively. Similarly, Figure 3 shows the DER values for nanoparticles of gold, platinum, iodine, silver, and iron oxide when using 220 kVp photon beams, displayed in Figure 3a–e, respectively. The depth along the central beam axis, as shown in Figure 1, was varied from 1 to 10 cm. It should be noted that kV photon beams have shorter depth doses than MV beams, with depth doses up to 20 cm, because kV photon beams are mainly used to treat skin lesions on the patient’s surface instead of deep-seated tumors [20,29]. Figure 2 and Figure 3 demonstrate that the addition of nanoparticles led to an increase in DER values, with a noticeable dose enhancement occurring within a depth of 1 cm. This is significant because most skin lesions have a depth within this range [30]. While the DER values decreased as the phantom depth increased, this may not be relevant because orthovoltage skin therapy is designed to treat skin cancer on the patient’s surface [31].

## 4. Discussion

### 4.1. Dependence of Dose Enhancement on the Phantom Depth

Phantom depth variations of DER values are demonstrated in Figure 2 and Figure 3. Figure 2a illustrates a decrease in DER value with increasing depth, particularly when nanoparticle concentration is higher. For instance, for a nanoparticle concentration of 40 mg/mL, the DER range was 3.77–0.32 within the 10 cm depth range, compared to a range of 1.10–0.49 for a concentration of 3 mg/mL. The observed variation in DER values can be attributed to two factors. Firstly, a higher concentration of nanoparticles leads to a greater dose enhancement and, consequently, a higher DER value. Secondly, the self-absorption effect of the phantom is more pronounced with higher nanoparticle concentrations due to the absorption of photons by nanoparticles along the depth of the phantom, leading to a greater reduction in DER values in the shallower depth range (0–5 cm) for higher nanoparticle concentrations [19]. However, for deeper depths (5–10 cm), the variation in DER values is negligible as most photons from the radiation beam have already been absorbed upstream. In orthovoltage skin therapy, where the tumor is situated on the surface of the patient’s skin, the 0–2 cm depth range is crucial for dose enhancement [11]. A similar trend in the variation of DER values with depth can be observed in Figure 2b–e and Figure 3a–e.

An intriguing finding regarding the variation of DER with depth is the presence of DER values that are close to or lower than one. This is particularly evident when the nanoparticle concentration is low, and the phantom is large. For instance, Figure 3a demonstrates that when the gold nanoparticle concentration is 3 mg/mL and the 220 kVp photon beam is used, the variation of DER values in the 10 cm depth range is within the range of 1.12–1.09. In comparison, using the 105 kVp beam in Figure 2a for similar DER results yields a variation range of 0.49–1.10. The underlying reason for such low DER values is that the introduction of heavy metal nanoparticles can act as a “shield” for the kV photon beam along the depth of the phantom, offsetting or even negating the dose enhancement effect [20]. This shielding or self-absorption effect is more prominent in Figure 2a due to the lower photon beam energy. This effect is more pronounced in deeper depths and lower nanoparticle concentrations. However, higher nanoparticle concentrations can still achieve a significant dose enhancement effect in skin therapy at shallow depth.

### 4.2. Dependence of Dose Enhancement on the Nanoparticle Material

The results of the study showed that the DER values increased with the atomic number of the nanoparticle material used. In Figure 2, which used a 105 kVp photon beam, gold nanoparticles exhibited the highest DER values among the materials tested. For example, the maximum DER values for a nanoparticle concentration of 40 mg/mL were 3.76 for gold, 3.68 for platinum, 3.27 for iodine, 3.37 for silver, and 1.53 for iron oxide nanoparticles. Similar DER results as shown in Figure 3 for the 220 kVp beam were 3.59, 3.56, 3.19, 2.86, and 1.27. Gold nanoparticles consistently exhibited the highest DER values at the surface of the phantom. In addition to their superior dose enhancement performance, gold nanoparticles are also easy to fabricate, biocompatible, and can be functionalized for various applications [32,33]. Platinum nanoparticles also showed good dose enhancement performance. However, iron oxide nanoparticles showed only a slight increase in DER values, likely due to their lower atomic number. It should be noted that in the kV energy range, photoelectric effect is dominant and its cross section is proportional to *Z*^n^, where *Z* is the atomic number, and *n* is between 4 and 5 [8]. In addition, it is important to note that iron oxide nanoparticles are magnetic and can enhance image contrast in magnetic resonance imaging, making them useful for image-guided radiotherapy [34,35].

### 4.3. Dependences of Dose Enhancement on Nanoparticle Concentration

In Figure 2 and Figure 3, with the same nanoparticle material and photon beam energy, increasing nanoparticle concentration led to an increase in DER values. This is due to the addition of more nanoparticles to the phantom, resulting in a higher compositional atomic number and, thus, more secondary electrons produced in the phantom. Gold nanoparticles in Figure 2a and Figure 3a showed a high DER value larger than 3 at the phantom surface with a nanoparticle concentration of 40 mg/mL. However, a low nanoparticle concentration of around 3 mg/mL resulted in a DER value close to 1. Similar trends were observed for platinum, iodine, silver, and iron oxide nanoparticles. For iron oxide nanoparticles in Figure 2e, DER values were lower than one at nanoparticle concentrations below 7 mg/mL due to the shielding effect being greater than the dose enhancement effect along the phantom depth. Therefore, a high nanoparticle concentration of over 30 mg/mL is recommended for adequate dose enhancement in orthovoltage nanoparticle-enhanced skin therapy.

Figure 2 and Figure 3 show that the impact of nanoparticle concentration on DER values diminishes as the phantom depth increases. In Figure 2a, the DER value increased from 1.10 to 3.77 (342%) at a depth of 0.5 cm as the nanoparticle concentration increased from 3 to 40 mg/mL. However, at a depth of 5.5 cm, the DER value only increased from 0.64 to 0.66 (3.1%). Similar results were observed for other nanoparticles in Figure 2 and Figure 3. This trend can be attributed to the fact that the dose enhancement effect due to the addition of nanoparticles is stronger near the phantom surface and is not affected by the self-absorption effect of the phantom. As a result, the variation of nanoparticle concentration truly reflects the dose enhancement effect at shallow depths. However, at deeper phantom depths, the self-absorption or shielding effect of the phantom with added nanoparticles becomes more significant, which weakens the dose enhancement effect. Therefore, the impact of nanoparticle concentration on DER values decreases with an increase in phantom depth.

### 4.4. Dependences of Dose Enhancement on the Photon Beam Energy

When comparing the DER values of gold nanoparticles at the same depth using 105 kVp (Figure 2a) and 220 kVp photon beams (Figure 3a), it can be observed that the maximum DER values were 3.77 and 3.59, respectively. The lower energy photon beam had a DER value that was 5.0% higher than that of the higher energy beam. This is due to the fact that in the kV energy range, the photoelectric cross section is proportional to 1/*E*^3.5^, where *E* represents the energy of the radiation beam [8]. For other nanoparticles, it was found that the DER values for platinum, iodine, silver, and iron oxide were 3.67%, 2.78%, 1.18%, and 1.17% higher, respectively, when the photon beam energy was changed from 220 kVp to 105 kVp. The lower values of the percentage increase can be attributed to the decrease in compositional atomic number within the phantom.

When comparing the maximum and minimum DER values at different depths in the phantom using 105 kVp and 220 kVp photon beams for gold nanoparticles (Figure 2a and Figure 3a), it can be observed that the deviations between the maximum and minimum DER values using the 105 kVp photon beam were 2.67 and 0.027 at depths of 0.5 and 5.5 cm, respectively. For the 220 kVp beam, the corresponding deviations were 2.38 and 0.677 at depths of 0.5 and 5.5 cm. The deviations in the maximum and minimum DER values due to nanoparticle concentration were larger for the 220 kVp photon beam. This is because the higher energy 220 kVp beam was more penetrative in the phantom, leading to the generation of more secondary electrons at deeper depths compared to the 105 kVp photon beam. A similar effect was observed for other nanoparticle materials.

## 5. Conclusions

Monte Carlo simulations were conducted to examine the depth dose enhancement in terms of DER in orthovoltage nanoparticle-enhanced skin therapy. The results indicate that gold nanoparticles outperformed other nanoparticles, such as platinum, iodine, silver, and iron oxide, in terms of dose enhancement at the phantom surface. Furthermore, the dose enhancement was found to increase with an increase in nanoparticle concentration and a decrease in photon beam energy. To achieve sufficient dose enhancement for treating skin lesions with the addition of nanoparticles, it is recommended to use a nanoparticle concentration of more than 30 mg/mL for treatment depths less than 2 cm. In order to delve deeper into the variations of dose enhancement and distribution in relation to nanoparticle variables, our future work will involve the utilization of human heterogeneous phantoms constructed from computed tomography image sets. This approach will enable us to conduct a more comprehensive investigation and gain insights into the impact of nanoparticle variables on the overall dose distribution.

## Figures and Tables

**Figure 1 micromachines-14-01230-f001:**
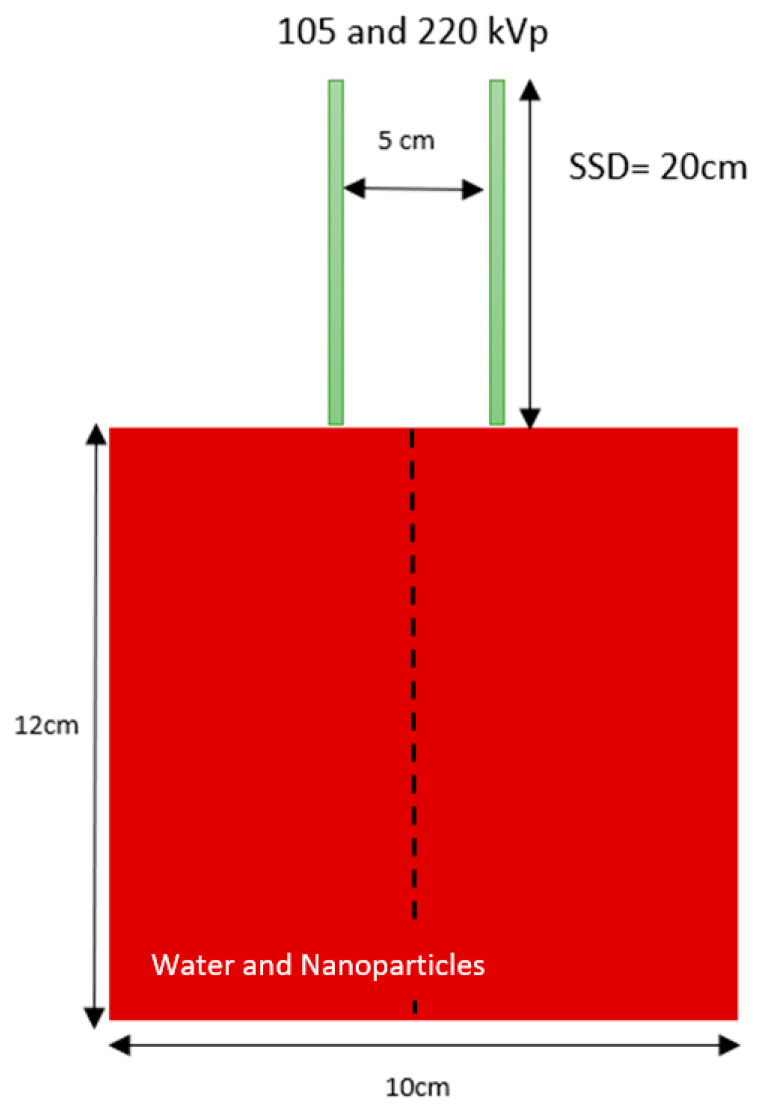
Schematic diagrams (not to scale) showing the heterogeneous phantom used in Monte Carlo simulations. The dimensions of the phantoms were equal to 12 × 10 × 10 cm^3^. The phantoms were irradiated by the 105 kVp and 220 kVp photon beams with field size equal to 5 cm diameter. The source-to-surface distance was equal to 200 cm.

**Figure 2 micromachines-14-01230-f002:**
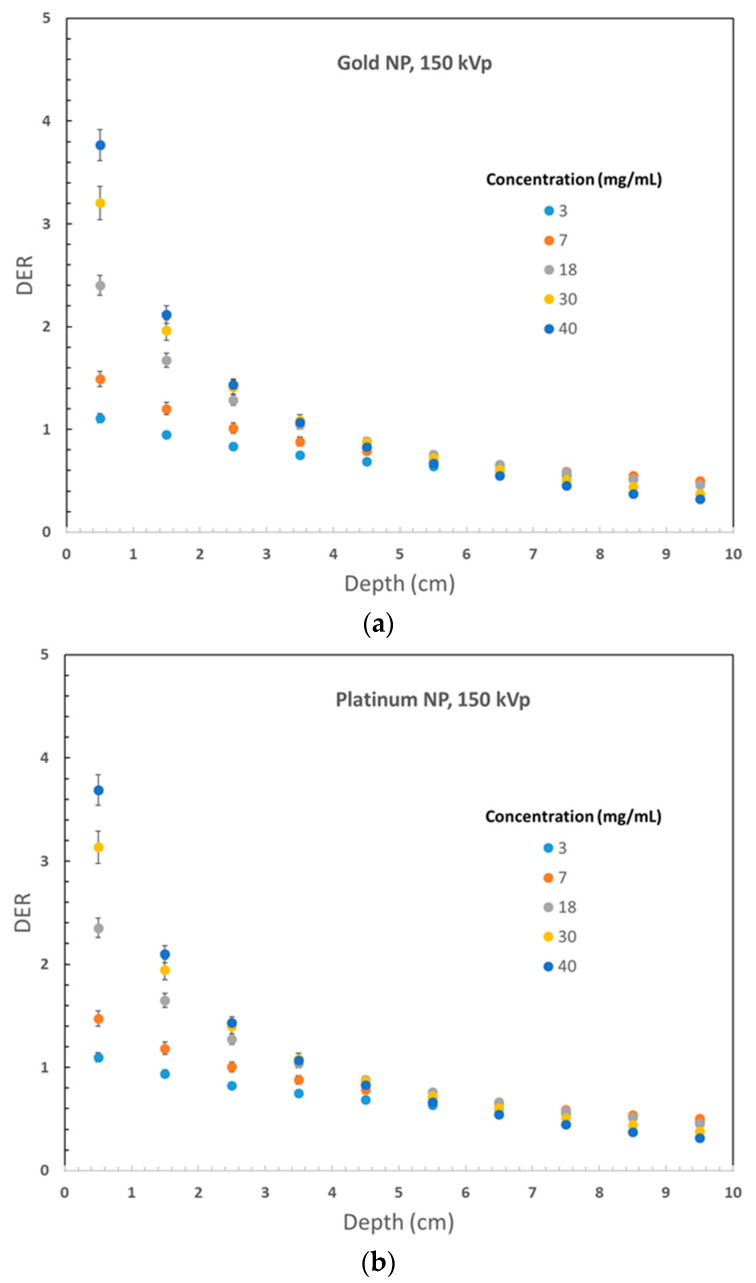
Relationships of dose enhancement ratio and phantom depth with variation of nanoparticle (NP) concentration using the 105 kVp photon beams for (**a**) gold, (**b**) platinum, (**c**) iodine, (**d**) silver, and (**e**) iron oxide.

**Figure 3 micromachines-14-01230-f003:**
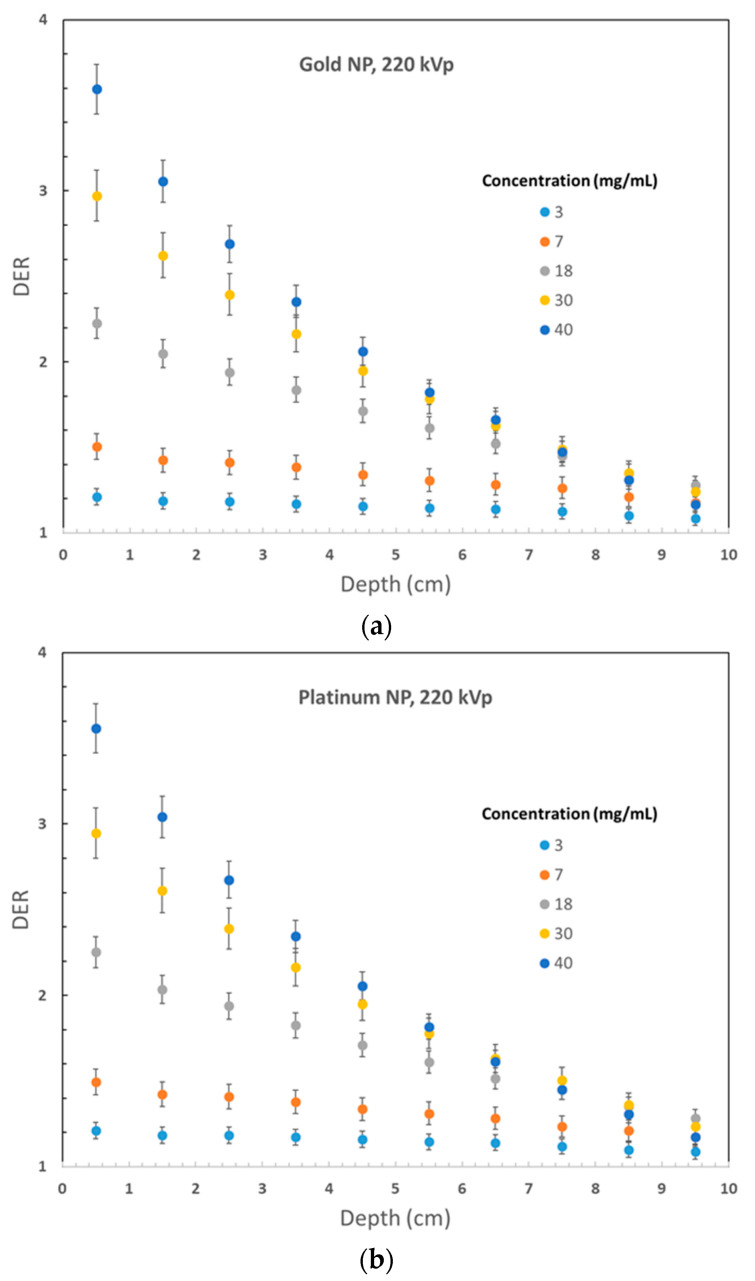
Relationships of dose enhancement ratio and phantom depth with variation of nanoparticle (NP) concentration using the 220 kVp photon beams for (**a**) gold, (**b**) platinum, (**c**) iodine, (**d**) silver, and (**e**) iron oxide.

## Data Availability

Not applicable.

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
