# Peer review of "Depth Dose Enhancement in Orthovoltage Nanoparticle-Enhanced Radiotherapy: A Monte Carlo Phantom Study"

_micromachines, 2023, doi:10.3390/mi14061230_

Round 1
Reviewer 1 Report
The paper is well written, the discussion is sound and the analysis of the results is complete, touching all the points of interest.
But my concerns are about the significance of the study. Although they have used a realistic energy spectrum of two X-ray machines, with appropriate reference to previous results, the work is just a simple simulation of particles in a water phantom with slightly modified material including the materials of the nanoparticles. This is a very simple simulation any novice simulation user can easily do, and therefore I do not see any scientific interest. The conclusions they reach are evident and do not need any simulation: the interactions increase with Z and decrease with photon energy, and there is a shielding effect when the previous traversed material produces more interactions.
The paper could be interesting if realistic cases were simulated: human phantoms (or at least a realistic skin) with nanoparticle concentrations similar to what can be obtained in a clinical procedure.
And one specific comment: as the study concentrates on skin effects, the DER vs depth analysis should provide a finer detail in that area (more data points at low depths)
Reviewer 2 Report
This paper present results obtained through Monte Carlo simulations of the dose enhancement ratio (DER) produced by nanoparticles of different materials in orthovoltage skin radiotherapy. The article is relatively well presented, the methodology is clear and the results are, overall, well presented. However, I have the following concerns:
Major issue:
1) The study is focused on the study of DER produced by nanoparticles in skin radiotherapy, however, the phantom doped with nanoparticles used in the simulations is unusually large (12x10x10 cm3) to represent healthy or tumor skin tissues. In this sense, there is another article already published (Zheng and Chow, 2017, doi: 10.4329/wjr.v9.i2.63) where this problem is addressed using a suitable phantom to represent the skin layer with nanoparticles. Moreover, in said article the DER was assessed for the same kilovoltage beams (105 and 220 kVp), same nanoparticles (Au, Pt, I, Ag and Fe2O3) at same concentrations (3, 7, 18, 30 and 40 mg/mL). In this regard, I consider that the purpose of this study should be reformulated, explaining in detail how it differs and what is the novelty of the results with respect to the article of Zheng and Chow, in order to it can be published.
Minor issues:
2) The generation of the X-ray beam used in the simulations is described between lines 94 and 101. However, it is not clear to me whether a phase space was used or if the X-ray tube and head of the Gulmay D3225 equipment were simulated. Furthermore, it is mentioned that 'The beam qualities of the 105 kVp and 220 kVp beams were defined by specific filters': How was this latter aspect accomplished? This entire part of the methodology needs to be better explained.
3) Lines 104 and 105: Why were the cutoff energies set to 512 keV for photons and 1 keV for electrons?
4) Line 109: Why were those concentrations chosen for the nanoparticles? Are these concentrations representative of the values used in radiotherapy treatments?
5) Lines 117 and 118: Why were 200 million histories simulated? What uncertainties were obtained for the dose estimators? What values of uncertainties are you aiming to achieve in this work? In this regard, uncertainty bands should be calculated and added to the graphs in Figures 2 and 3.
Reviewer 3 Report
Dear Authors
The paper is well written; authors should pay attention to the following points:
1-In Materials and Methods, no explanation was given on how the nanoparticles were simulated.
2-Was the distribution of nanoparticles homogeneous?
Regards
Round 2
Reviewer 2 Report
The manuscript has been improved to the point that, in its present form, it is suitable for publication.
Author Response
Thank you so much for your comments improving this work.